# A Highly Selective Turn-On Fluorescent Probe for the Detection of Zinc

**DOI:** 10.3390/molecules26133825

**Published:** 2021-06-23

**Authors:** Ling-Yi Shen, Xiao-Li Chen, Xian-Jiong Yang, Hong Xu, Ya-Li Huang, Xing Zhang, Carl Redshaw, Qi-Long Zhang

**Affiliations:** 1The Key Laboratory of Environmental Pollution Monitoring and Disease Control, School of Public Health, Ministry of Education, Guizhou Medical University, Guiyang 550014, China; shenly@stumail.nwu.edu.cn; 2School of Basic Medical Science, Guizhou Medical University, Guiyang 550004, China; C13885297112@163.com (X.-L.C.); xuhong@gmc.edu.cn (H.X.); ylh6401@gmc.edu.cn (Y.-L.H.); zhangxing11207115@126.com (X.Z.); 3Department of Chemistry, University of Hull, Cottingham Road, Hull, Yorkshire HU6 7RX, UK; c.redshaw@hull.ac.uk

**Keywords:** fluorescent probe, Schiff base, crystal structure, titration experiments

## Abstract

A novel turn-on fluorescence probe **L** has been designed that exhibits high selectivity and sensitivity with a detection limit of 9.53 × 10^−8^ mol/L for the quantification of Zn^2+^. ^1^H-NMR spectroscopy and single crystal X-ray diffraction analysis revealed the unsymmetrical nature of the structure of the Schiff base probe **L**. An emission titration experiment in the presence of different molar fractions of Zn^2+^ was used to perform a Job’s plot analysis. The results showed that the stoichiometric ratio of the complex formed by **L** and Zn^2+^ was 1:1. Moreover, the molecular structure of the mononuclear Cu complex reveals one ligand **L** coordinates with one Cu atom in the asymmetric unit. On adding CuCl_2_ to the ZnCl_2_/**L** system, a Cu-Zn complex was formed and a strong quenching behavior was observed, which inferred that the Cu^2+^ displaced Zn^2+^ to coordinate with the imine nitrogen atoms and hydroxyl oxygen atoms of probe **L**.

## 1. Introduction

Zinc is the second most abundant transition metal ion in the human body and is essential for various biochemical processes, such as neurotransmission, enzyme regulation, gene expression, and apoptosis [1,2,3,4]. In addition, zinc at normal concentrations controls many metabolic, biological, and environmental processes, while deficiency usually leads to the appearance of some clinical diseases, such as growth retardation, brain dysfunction, high blood cholesterol, Parkinson’s disease, ischemic stroke, Alzheimer’s disease, etc. [5,6,7,8,9]. On the other hand, excessive zinc will also cause problems for humans; for example, it will reduce soil microbial activity and cause phytotoxic effects [10,11,12,13,14]. Therefore, it is of great significance to be able to accurately detect zinc ion concentration [15,16,17,18].

At present, the main detection methods of zinc include spectrophotometry, electrochemical methods [19], atomic absorption methods [20], chromatographic methods, and mass spectrometry [21,22]. However, due to the high cost of equipment, cumbersome sample preparation, or prolonged testing time, the wide application of these methods in actual testing has been limited. Among various detection methods, metal ion fluorescent chemical sensors have attracted much attention because of the convenient use, high sensitivity, and ability to directly measure concentration through fluorescent signals [23,24,25,26,27]. Zn^2+^ fluorescent probes can be divided into three categories: fluorescence quenching, fluorescence enhanced type, and ratio type. There is still an urgent need to develop new small molecule probes that are easy to prepare, have high sensitivity, and can recognize Zn^2+^ with excellent selectivity.

Schiff bases can coordinate with a variety of metals due to their special chemical structure, so they are considered to be dominant ligands for metal ions [28]. Schiff bases and their transition metal complexes not only have a wide range of applications in synthesis, catalytic chemistry, and materials chemistry [29], but also have a wide range of applications in antibacterial, antifungal, anticancer [30], clinical, analytical, and pharmacological aspects [31,32,33,34]. Moreover, a series of Schiff base ligands have been reported to be fluorescence turn-on chemosensors for Zn^2+^ with high sensitivity and selectivity [35,36,37,38,39,40]. Such systems are based on several reported mechanisms of fluorescence enhancement behavior, including internal charge transfer (ICT) [41], chelating-enhanced fluorescence (CHEF) [42,43], photoinduced electron transfer (PET) [44,45], aggregation-induced emission (AIE) [46,47,48], and C=N isomerization mechanisms [49,50,51]. In the current work, we have designed an asymmetric Schiff base **L** that not only has the ability to chelate metals but also has lone pair electrons on nitrogen atoms. The synthesis of **L** is shown in Scheme 1, and its structure has been confirmed by NMR spectroscopy, mass spectrometry, single crystal X-ray diffraction, and UV-Vis spectroscopy.

The fluorescence spectrum indicated little fluorescence emission of the free probe **L**. However, after adding Zn^2+^, **L** exhibited a fluorescence emission peak at 475 nm and a 512-fold fluorescence enhancement. Meanwhile, qualitative and quantitative detection are achieved through the linear relationship between fluorescence intensity and zinc ion concentration. Additionally, most of the coexisting metal ions had little or negligible interference on the emission response of probe **L** toward Zn^2+^. Of particular note is Cd^2+^, which has very similar chemical properties to Zn^2+^, and most Zn^2+^ sensors tend to respond to both Zn^2+^ and Cd^2+^ [32,33,34,35]. Hence, the development of a Zn^2+^ selective fluorescence sensor that can discriminate Zn^2+^ from Cd^2+^ is a great challenge and is of great significance. The recognition mechanism of the probe for Zn^2+^ in ethanol solution is proposed to be C=N isomerization [49,50,51] and chelation-enhanced fluorescence [42,43]. The C=N isomerization is inhibited by the coordination of Zn^2+^ with the probe **L**, so the fluorescence is significantly enhanced. Moreover, both Zn^2+^ and Cu^2+^ were investigated as the metal ions to coordinate simultaneously with **L** in order to understand the strong quenching behavior of Cu^2+^.

## 2. Results and Discussion

### 2.1. Selectivity of the Probe ***L***

To determine potential practical applications, the spectroscopic properties of **L** were measured under simulated physiological conditions (50 μM in ethanol solution). The probe will be deactivated when the water content exceeds 5%, and thus the fluorescence spectral response of **L** to metal ions was recorded in ethanol solution excited at 354 nm as shown in Figure 1. The fluorescence spectroscopic response of **L** toward metal ions was evaluated in ethanol solution upon excitation at 354 nm as presented in Figure 1. The fluorescence spectroscopy indicated that the addition of Zn^2+^ resulted in a significant enhancement of the emission intensity positioned at 475 nm (Figure 1A, blue line). Under the same conditions, next to no responsive changes are observed in the presence of 1 equiv. of various metal cations (solutions of Zn^2+^, Li^+^, Na^+^, K^+^, Ag^+^, Mg^2+^, Ca^2+^, Sr^2+^, Ba^2+^, Al^3+^, Fe^3+^, Co^2+^, Ni^2+^, Cu^2+^, Pb^2+^, Cd^2+^, and Hg^2+^ were prepared from their chloride salts) in ethanol. Thus, according to the spectroscopy changes, the Schiff base probe **L** can detect Zn^2+^ with good selectivity.

### 2.2. Competition Experiments

It is necessary for a metal ion fluorescence chemosensor to achieve higher selectivity over other competing metal ions. Therefore, we evaluated the fluorescence behavior of probe **L** for Zn^2+^ in the presence of various competing metal ions, in which **L** was treated with 1 equiv. of Zn^2+^ in the presence of 1 equiv. of other metal ions. Figure 1B summarizes the results, and reveals that the presence of Li^+^, Na^+^, K^+^, Ag^+^, Hg^2+^, Mg^2+^, Hg^2+^, Ba^2+^, and Pb^2+^ caused only minor interference for the detection of Zn^2+^, whilst the presence of Co^2+^, Al^3+^, Ca^2+^, Ni^2+^, and Cr^3+^ resulted in low fluorescence intensity but were clearly detectable. However, in the case of Fe^3+^ and Cu^2+^, quenching of the fluorescence signal was observed, which may be due to the paramagnetic nature of Cu(II) and Fe(III) and their stronger coordinating ability than Zn(II). The fluorescence probe for detecting Zn^2+^ in the presence of coexisting metal ions, except for Fe^3+^ and Cu^2+^, clearly maintains higher emission than observed for the free probe **L**. Moreover, Cd^2+^ did not inhibit the emission intensity of Zn^2+^. Thus, these results indicate that probe **L** could be used as a selective probe for Zn^2+^ for distinguishing Zn^2+^ from Cd^2+^, which commonly share similar properties.

### 2.3. Fluorescence Spectroscopic Studies of ***L*** toward Zn^2+^

The result of the fluorometric titration of the free probe **L** and those in the presence of incremental amounts of Zn^2+^ in ethanol solution is shown in Figure 2A. The fluorescence spectrum showed that the fluorescence intensity increases steadily and smoothly on increasing the Zn^2+^ concentration. On addition of up to 1 equiv. of Zn^2+^, the turn-on ratio was observed to increase by over 512-fold. Additionally, the dependence of the emission intensity at 475 nm on the Zn^2+^ concentration is shown in Figure 2B. The fluorescence intensity remained constant in the presence of more than 1 equiv. of Zn^2+^, and therefore, the formation of a 1:1 complex between **L** and the Zn^2+^ was proposed. According to the fluorescence titration data, the association constant for **L**–Zn^2+^ complexation was calculated at 1.42 × 10^4^ mol/L from the Benesi-Hildebrand plot (Figure 2C). For practical purposes, the detection limit of probe L is an important parameter. The detection limit of probe **L** for Zn^2+^ was determined to be 9.53 × 10^−8^ mol/L according to the IUPAC definition (CDL = 3 Sb/m) from 10 blank solutions [52,53]. The Job’s plot analysis shows that a maximum emission was observed when the molar fraction reached 0.5, suggesting that the complex formation between **L** and Zn^2+^ has the stoichiometric ratio of 1:1. A linear relationship (R^2^ = 0.99173) for the plot of the normalized fluorescence intensity at 475 nm against Zn^2+^/**L** is shown in Figure 2D. 

For a rough comparison between present complex with other reported similar fluorescent sensors [35,37,38,39,54,55], based on the detection limits of fluorescent sensors for Zn^2+^, see the results gathered in Table 1. The present compound is clearly more sensitive than most of the other Schiff base fluorescent sensors for the detection of Zn^2+^ via turn-on fluorescence.

### 2.4. UV-Vis Absorbance Response of Probe ***L*** towards Zn^2+^

As illustrated in Figure 3, the probe **L** exhibited a maximal absorption at 313 nm. Upon addition of Zn^2+^ ions (0−1 equiv.), the absorbance at 313 nm decreased gradually on gradually increasing the Zn^2+^ concentration (Appendix A, Appendix A). The presence of two clear isosbestic points at 273, 339 nm is consistent with the conversion of the free probe **L** to the Zn^2+^ complex. Moreover, the absorbance at 313 nm hardly changes in the presence of more than 1 equiv. of Zn^2+^ ions, indicating the formation of a 1:1 complex between **L** and the zinc ion. This is in good agreement with a 1:1 stoichiometry for the Zn^2+^ complex as determined by the Job’s plot obtained from UV-Vis absorption (Appendix A, Appendix A).

### 2.5. Crystal Structures of Probe ***L*** and Metal Complexes

To further investigate the binding mode of **L** with other metal ions, probe **L** and the organic framework of **L** possess an approximate torsional structure which chelates with one equivalent of Cu^2+^ ions. Moreover, a heteronuclear bimetallic complex was also obtained. The geometrical parameters of the mononuclear Cu complex, Cu-Zn complex, and **L** are listed in Appendix A, Appendix A. Treatment of 2,4-pentanedione with (1*R*, 2*R*)-diaminocyclohexane in refluxing methanol for 15 min, cooling, followed by the addition of 2-hydroxy-3-methoxybenzaldehyde led to the target product **L**, which was obtained from the filtrate on standing overnight below 0 °C. The reaction generally affords a high yield (84%), and yellow crystals suitable for X-ray determination were obtained (Figure 4a). The probe **L** is found to be unsymmetrical as shown by the crystal structure and the potential coordination sites in the two side chains point in opposite directions. Additionally, the bond lengths of N1–C17 and N2–C7 are 1.319(10) Å and 1.310(12) Å, respectively, corresponding to a C=N double bond. Both bond lengths of C15–C16 (1.348(15) Å) and C15–O1 (1.293(14) Å) were between single and double as a result of contributing to the conjugation effect. Moreover, all the atoms (except for hydrogen) in each side chain are coplanar and the dihedral angle of the two planes is 61.28°, thereby making the whole molecule appear “V” shaped.

To synthesize the copper complex, ligand **L** was treated with an equivalent of CuCl_2_ at room temperature, and following work-up, brown crystals were afforded. X-ray crystallographic analysis (Figure 4b) shows that the Cu^2+^ complex is a mononuclear complex, where the asymmetric unit consists of one ligand **L** and one Cu atom, giving rise to a 1:1 ligand to metal coordination. The central Cu^2+^ is four-coordinate with one phenoxide oxygen atom, and one enolic hydroxyl oxygen atom and two imine nitrogen atoms. Moreover, the four-coordinated atoms are almost coplanar, in which the dihedral angle of N1–Cu1–O1 and N2–Cu2–O2 is only 9.21°. The bond lengths of N1–C17 and N2–C7 are 1.309(5) Å and 1.290(5) Å, respectively, and are shorter than in the neutral ligand. The bond lengths to Cu are comparable with corresponding values observed in similar complexes [56,57]. The Cu complex clearly shows quenched fluorescence, which might be due to the d^9^ electron configuration of the Cu^2+^ ions making the transfer of ligand electrons from the excited states to the d-orbital of Cu^2+^ rather than transferring back to the ground state of the ligand [58].

In order to understand the strong quenching behavior when coexisting Cu^2+^ and Zn^2+^ are present, we added CuCl_2_ to the solution of ZnCl_2_/**L**. As expected, the solution shows an obvious quenched fluorescence. Moreover, a heteronuclear bimetallic complex was obtained by concentrating the solution. The solid-state molecular structure is shown in Figure 5, as determined by X-ray crystallography. This revealed that the asymmetric unit is composed of two fragments that are connected by two bridged chlorine atoms to construct a dimeric heteronuclear metal complex. The coordination environment of the copper atom is a slightly distorted square-planar involving two O and N atoms from the **L**^2−^ ligand, which is similar to that in the above-mentioned mononuclear Cu-complex. Moreover, the zinc has a distorted tetrahedral environment, in which the zinc atom is coordinated by the two O atoms of the **L**^2−^ ligand and the two Cl atoms. The average angle of Cl–Zn–Cl is 116.96(11) and the Cl–Zn–Cl planes are nearly perpendicular to the Cu**L** units, respectively, which decreases the steric effects between them. The distances between Cu and Zn are 3.106 Å and 3.101 Å in each half of the molecule, respectively. The core feature of the bimetallic structure possesses an average Cu–O bond of 1.919 Å and Zn–O bond of 2.146 Å, accompanied by an average Zn–Cl bond of 2.220 Å and Cu–N bond of 1.939 Å. These values are comparable with corresponding values observed in similar Cu-Zn complexes [59,60]. The formation of the Cu-Zn complex indicates that the Cu^2+^ can displace the Zn^2+^ to coordinate with the imine nitrogen atoms and the hydroxyl oxygen atoms of probe **L**. In this regard, the formation of complex Cu-Zn would be the main cause of the strong quenching behavior when Cu^2+^ coexists in the presence of Zn^2+^/**L**.

### 2.6. Proposed Recognition Mechanism

To better understand the recognition mechanism in the detection of **L** for Zn^2+^, the ^1^H NMR spectroscopic titration experiment was performed as shown in Appendix A, Appendix A. It shows that the proton peaks at δ = 13.49 and 10.74 ppm of the phenolic hydroxyl and enol hydroxyl, respectively, of probe **L** gradually decreased on the addition of zinc ions. Especially, the proton peak of the phenolic hydroxyl was almost completely gone when the addition of Zn^2+^ reached 1.0 equivalent, which indicated that the Zn^2+^ coordinates to the two O atoms with the stoichiometric ratio 1:1. The enol hydroxyl proton peak remains because the enol and ketone can still interchange. On the other hand, based on the Job’s plot analysis and the structures of similar types of zinc complexes reported in the literature [61,62,63], we propose the structure of a 1:1 complex for **L** and Zn^2+^ is as shown in Scheme 2. The remarkable increase of the fluorescence of probe **L** at 475 nm can be explained as follows: the probe **L** with a C=N containing structure shows little fluorescence because of C=N isomerization, which leads to the predominant decay in the excited states [64]. In contrast, the nitrogen containing lone pair electrons coordinate with the zinc ions and form a bond to restrain the C=N isomerization so that its fluorescence increases drastically [65]. Moreover, the complexation of Zn^2+^ with **L** leads to a more rigid molecule, and produces a large chelation-enhanced fluorescence detection effect, which leads to a large increase in the fluorescence [66]. On the other hand, it has been reported that transition metal cations with closed shell d-orbitals cannot form low-energy metal center excited states or charge-separated excited states to provide the obvious fluorescence enhancement. However, transition metal cations with open shell d-orbitals usually quench fluorescence due to electron or energy transfer between the metal cation and the fluorophore, providing rapid and effective non-radiative decay of the excited state [67]. The formation of Cu-Zn complex implies that Cu^2+^ with its open shell d-orbitals replaces Zn^2+^ with closed shell d-orbitals and coordinates with C=N bond of the ligand. Hence, strong quenching of the fluorescence was observed following the addition of CuCl_2_ to the ZnCl_2_/**L** system.

## 3. Materials and Methods

### 3.1. Reagents and Equipment

All of the starting materials and solvents were commercially available and used without further purification. Ultrapure water was used throughout the experiments. The solutions of the metal ions were prepared from their chloride salts. The UV-Vis absorption spectra were determined at room temperature on a UV-2600 Milton Ray Spectrofluorometer (Shimadzu, Kyoto, Japan) in a 1 cm quartz cell. Fluorescence spectroscopy measurements were recorded on a Cary Eclipse Hitachi 4500 spectrophotometer (Hitachi, Tokyo, Japan) (Varian). ^1^H-NMR spectra were measured using an Inova-600 Bruker AV 600 spectrometer (Bruker, Karlsruhe, Germany) at room temperature. DMSO-*d*_6_ was used as a solvent and tetramethylsilane (TMS) as an internal standard. Single crystal X-ray diffraction was conducted on a Bruker Smart Apex II single crystal diffractometer (Bruker, Karlsruhe, Germany). The spectroscopic properties of the probe **L** were investigated in an ethanol solution at 1 mmol/L that was diluted to the required concentrations. All the metal ions and anions were initially prepared as a 1 mmol/L ethanol solution and then diluted to the needed concentrations.

### 3.2. Synthesis of the Fluorescent Probe ***L***

Here, 2,4-pentanedione (1.00 g, 10 mmol) in methanol (15 mL) was added dropwise to a methanol (20 mL) solution of (1*R*, 2*R*)-diaminocyclohexane (1.14 g, 10 mmol). The mixture was heated to reflux for 15 min, and cooled to room temperature. Then, 2-hydroxy-3-methoxybenzaldehyde (1.52 g, 10 mmol) was added and the solution was stirred for an additional 2 h at room temperature. It was filtered and the filtrate was left to stand overnight below 0 °C. The resulting yellow precipitate was collected by filtration to give the Schiff-base ligand **L** (2.67 g, 84%). The precipitate was recrystallized from methanol. X-ray quality yellow single crystals of ligand **L** were obtained by slow evaporation of a saturated methanol solution. ^1^H-NMR (600 MHz, DMSO-TMS): δ/ppm 13.49 (s, 1H, Ar–OH), 10.74 (s, 1H, (C=C)OH), 8.39 (s, 1H, CH=N), 7.03 (d, *J* = 7.8 Hz, 2H, Ar–H), 6.94 (d, *J* = 7.8 Hz, 2H, Ar–H), 6.80 (t, *J* = 7.8 Hz, 1H, Ar–H), 4.74 (s, 1H, CH=C), 3.77 (s, 3H, CH_3_–O), 3.61 (m, 1H, Cy–H–N), 3.10 (m, 1H, Cy–H–N), 1.81 (s, 3H, CH_3_(C=N)), 1.78 (s, 3H, CH_3_(C=C)), 1.95−1.37 (m, 8H, Cy−CH_2_). ^1^^3^C-NMR (150.9 MHz, DMSO-TMS): δ/ppm 193.2 (C=C–O*H*), 165.6 ( CH_3_–C=N), 162.3 (Ar–C=N), 151.0 (Ar–C–O*H*), 148.0 (Ar–C–C*H*_3_), 123.2 (Ar–C–(C=N)), 118.4 (Ar–C), 118.1 (Ar–C), 114.8 (Ar–C), 94.7 (C=C–O*H*), 72.5 (Cy–C–N), 55.9 (Cy–C–N), 55.8 (CH_3_–O), 32.9, 32.6, 28.6, 24.2 (Cy–CH_2_), 23.5 (CH_3_(C=C)), 18.8 (CH_3_(C=N)). ESI-MS *m*/*z*: calcd. for [C_19_H_26_N_2_O_3_ + H]^+^, 331.2016; found, 331.2011. Elemental analysis calcd. (%) for C_19_H_26_N_2_O_3_ [330.19 g/mol]: C 68.85, H 8.21, N 8.45. Found (%): C 68.56, H 8.47, N 8.73.

### 3.3. X-ray Crystallography

Diffraction data for the probe **L**, Cu-complex, and the Zn-Cu complex were collected on a Bruker SMART APEX II diffractometer at room temperature (298 K) with graphite-monochromated Mo Kα radiation (λ = 0.71073 Å). An empirical absorption correction using SADABS was applied for all data [68]. The structures were solved and refined to convergence on F2 for all independent reflections by the full-matrix least squares method using the SHELXL−2014 programs [69] and OLEX2 1.2 [70]. Hydrogen atoms bonded to carbons were included in idealized geometric positions with thermal parameters equivalent to 1.2 times those of the atom to which they were attached. In compound **L**, one oxygen atom in the solvent water was disordered and there was a large amount of disorder in the structure. In particular, the disordered side-chains are very dynamic and may be considered as a solvent. Short contacts between disordered fragments are to be expected, which caused the observed level B alerts. Crystallographic data and refinement details for **L**, Cu-complex, and the Zn-Cu complex are given in Appendix A, Appendix A. CCDC: 2067620, **L**; 2067621, Cu-complex; and 2067622, Zn-Cu complex contain the supplementary crystallographic data for this paper. These data can be obtained free of charge from the Cambridge Crystallographic Data Centre: www.ccdc.cam.ac.uk/data_request/cif.

### 3.4. General Procedure for Analysis

Before conducting the spectroscopic measurements, the corresponding solutions of probe **L** and the metal ions were freshly prepared. For fluorescence spectroscopy selective experiments, test solutions were prepared as follows: a stock solution of probe **L** (1 mM) and the reactive species (1 mM) was prepared in ethanol solution and 0.15 mL of probe **L** and 0.15 mL of the reactive species were placed into a 3 mL cuvette and then after diluting the solution to 3 mL with ethanol solution; the final concentration was 50 μM. The fluorescence spectra were collected at room temperature.

For fluorescence spectroscopy competitive experiments, test solutions were prepared as follows: a stock solution of probe **L** (1 mM) and the reactive species (1 mM) was prepared in ethanol solution and 0.15 mL of probe **L** and 0.15 mL of the reactive species were placed into 3 mL cuvette and mixed. Next, 0.15 mL of the Zn^2+^ solution was added to the above mixed solution, which was then diluted to 3 mL with ethanol. Fluorescence spectra were recorded at room temperature. 

For UV-Vis and fluorescence spectroscopy titrations experiments, a stock solution of the probe **L** (1 mM) was prepared and 0.15 mL of the probe **L** solution was placed into 3 mL cuvette. Adding 15–450 µL Zn^2+^ solution to the above cuvette, and then diluting the solution to 3 mL with different amounts of ethanol solution. UV-Vis and fluorescence spectra were taken at room temperature. 

For the Job’s plot measurement, the total concentration of the probe **L** and Zn^2+^ was kept at 50 µmol. **L** and the ratio of the concentration of Zn^2+^ to the concentration of the probe **L** was changed to 0:10, 1:9, 2:8, 3:7, 4:6, 5:5, 6:4, 7:3, 8:2, 9:1, and 10:0, with the fluorescence intensity at 475 nm and the ultraviolet absorption intensity at 310 nm as the vertical axis Zn^2+^ occupies the probe and the molar ratio of Zn^2+^ is on the horizontal axis. The spectra of these solutions were immediately recorded by means of the UV-Vis method and fluorescence spectroscopy.

## 4. Conclusions

In summary, a new, low-cost, rapid, and portable Schiff base fluorescent probe **L** was synthesized and characterized. A “turn-on” fluorescence emission was observed upon sequential addition of Zn^2+^ and this increases proportionally with increased Zn^2+^ concentration. Meanwhile, it was found that probe **L** has great fluorescence selectivity for Zn^2+^ over many other important metal ions. Additionally, the method of equivalent molarity of fluorescence and UV spectrum indicated a 1:1 binding mode between **L** and Zn^2+^. Finally, the formation of the Cu-Zn complex and the strong quenching behavior of coexisting Cu^2+^ for Zn^2+^ may lead to the potential application as an on-off chemosensor candidate for Cu^2+^.

## Data Availability

No new data were created or analyzed in this study. Data sharing is not applicable to this article.

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
