# Peer review of "A Highly Selective Turn-On Fluorescent Probe for the Detection of Zinc"

_molecules, 2021, doi:10.3390/molecules26133825_

Round 1

Reviewer 1 Report

This report for a Zn2+ sensor could be of interest for the community developing chemosensors and it is appropriate for Molecules. However, some improvements are necessary to make clear the scientific contribution. 

1) Did authors try to test the ligand in aqueous solutions. The contribution would be more significant if authors report the performance of the ligand in a mixture of ethanol +  water. Notice that the relevant sensing must be in water not in ethanol. In case the ligan is not stable or not soluble in water discuss this limitation. The ligand should work in mixture with at least a minimum of water

2) Conclusions should include a comparative table this chemosensor with some of the great variety of fluorescent sensors reported for Zn2+ or similar ions.

Reviewer 2 Report

The manuscript describes new Shiff base probe for zinc detection. Good selectivity for zinc is reported, however, the experiments were run in ethanol. Not many researchers are interested in zinc detection in methanolic solutions and, thus, only very low interest of readers could be expected. Further, the ligand characterization is insufficient and competition with other metal ions should be studied more in details to understand the system. Therefore, a very major revision is required.

Comments:

  1. The introduction is insufficient. There is no explanation of the ligand design. No structurally similar ligands and probes are discussed. Referencing in the introduction must be improved – most of the introduction are general statements, but corresponding references (1-17) are mostly randomly selected original research articles. Such statements should be referenced mostly with reviews and book chapters.

  1. Ligand characterization is insufficient. 13C NMR, MS and elemental analysis are missing. 1H NMR spectra should be scaled to show ligand signals in appropriate intensity. Purity of the ligand is not good according to 1H NMR spectra, signals of impurities are observed in regions 1-2 ppm and 3-4 ppm.

  1. Zinc detection is mostly related to biomedical application and, thus, probes working in aqueous solutions are of interest. Thus, pilot experiments in aqueous solutions should be done and reported.

  1. The detection limit should be presented with not more than one decimal.

  1. What is reason for fluorescence quenching with calcium? Zinc cannot be replaced with calcium in the complex. And formation of a ternary complex with Ca is not reasonable as other (better complexing) metal do not show such quenching.

  1. Slope of the plot in Figure 2B in the range 0-1 does not agree with the zero fluorescence intensity of free ligand. Further, the data in Figures 2B and 2C do not correspond to each other. In addition, the Job plot is redundant as the metal-to-ligand ration could be clearly identified in the experiment shown in the Figure 2b.

  1. The dimer structure in Figure 5 looks to be bridged by bonds between chlorine and oxygen. Also graphicals presentation is not good. A better figure should be shown.

Reviewer 3 Report

The manuscript submitted by Qi-Long Zhang and co-authors explained the synthesis of unsymmetrically substituted Schiff base ligand for detection of Zn2+ ions and the effect on addition of Cu2+ ions along with single crystal X-ray diffraction analysis. Although Zn2+ fluorescent sensor is very common, however, dimeric crystal structure of bimetallic Cu2+-Zn2+ complex is an important aspect. Thus, the manuscript may be suitable for publication, but following major comments should be addressed.

  1. Authors should incorporate 1H NMR titration of Schiff base L with Zn2+ ions to record the interactions since they don’t have the crystal structure with it.
  2. Although ligand L shows selective “turn-on fluorescence” towards Zn2+ ions, it also strongly interacts with Cu2+ ions which even can displace Zn2+ ions from the coordination site. Similar kind of interactions are expected with other metal ions (Al3+, Ni2+, Ca2+, Cr3+, Fe3+) since compound L did not show any emission enhancement with Zn2+ ions in presence of these metal ions. Hence, authors need to record competitive 1H NMR of L with Zn2+ ions in presence of above mentioned metal ions (except paramagnetic Cu(II) and Fe(III) ions). Authors also need to cite these articles where compounds with same binding units forms complexes with Cu2+, Ni2+ and Fe3+ ions (Tetrahedron: Asymmetry 1998, 9, 3741–3744 and Chinese Journal of Inorganic Chemistry, 2012, 28(2), 321-325).
  3. Authors mentioned in manuscript the 1H NMR spectra of ligand L shows singlet at 10.73 (s, 1H, OH) while in spectra it shows doublet. They need to explain.

Moreover, they have written there is doublets of doublet at 7.03−6.92 (dd, 2H, Ar-H), authors need to consider coupling constant values and it seems like there are two separate doublets (d) instead of doublets of doublet (dd).

Authors also should incorporate 13C and mass of ligand in support of characterization.

  1. Which salts of other metal ions have been used in selectivity fluorescence experiment depicted in Figure 1?
  2. Authors also need to discuss Zn(II) detection based on aggregation-induced emission in the introduction (cite some articles for instance: Journal of Luminescence, 2018, 194 366-373; New J. Chem., 2017, 41, 4806-4813 and J. Mater. Chem. C, 2017, 5, 9651-9658).
  3. They should mention the excitation wavelength at figure captions of fluorescence spectra.
  4. In Figure 2a-b, fluorescence intensity increased by huge extent (around 170 unit) upon first addition of Zn2+ ions, how much metal ions have been added to make a rapid increment in intensity?
  5. Authors need to explain or clarify the alert B in crystals shown in checkcif.
  6. Authors got dimeric bimetallic crystal upon addition of CuCl2 to the complex of L/ZnCl2. So, it would be interesting to include what happen if ZnCl2 has been added to the complex of L/CuCl2.

Round 2

Reviewer 1 Report

In my opinion the manuscript can be accepted for publication provided author include a couple of lines decribing or justifying the use of ethanol instead of aqueous solutions. If the ligand is not longer sensitive to zinc in aqueous solution with low (5%) contect of water, this fcat must be clear for readers.

Author Response

Q1-1. In my opinion the manuscript can be accepted for publication provided author include a couple of lines decribing or justifying the use of ethanol instead of aqueous solutions. If the ligand is not longer sensitive to zinc in aqueous solution with low (5%) contect of water, this fcat must be clear for readers.

A1-1: Thank you very much for your comments. We have added to the introduction that we use ethanol as the solvent because the probe is not able to recognize zinc ions when the water content exceeds 5%.

Reviewer 2 Report

The manuscript has been improved and all quastions were answered. Thus, I recommend to publish in the present form.

Author Response

Reviewer: 2  “The manuscript has been improved and all quastions were answered. Thus, I recommend to publish in the present form.”

A1: Thank you very much for your affirmation.

Reviewer 3 Report

Qi-Long Zhang and co-authors have revised the manuscript precisely and now it can be accepted for publication in Molecules. But minor comments are required to address before publishing it in this journal.

  1. Authors should incorporate a discussion in the manuscript about the 1H NMR titration of probe upon addition of ZnCl2 shown in Fig. S3.
  2. They need to correct the citation of “Fig. S3” in manuscript since the discussion for Job’s plot depicted in “Fig. S6”.

Author Response

Q3-1.Authors should incorporate a discussion in the manuscript about the 1H NMR titration of probe upon addition of ZnCl2 shown in Fig. S3.

A3-1: We have added a discussion to the manuscript about the 1H NMR titration of probe upon addition of ZnCl2 as shown in Figure S3. (in Proposed recognition mechanism section, around lines 306-313, marked in blue)

Q3-2.They need to correct the citation of “Fig. S3” in manuscript since the discussion for Job’s plot depicted in “Fig. S6”.

A3-2: We have corrected the citation of “Fig. S3” to “Figure S6”.